# Antibacterial Activity of Juglone Revealed in a Wound Model of *Staphylococcus aureus* Infection

**DOI:** 10.3390/ijms24043931

**Published:** 2023-02-15

**Authors:** Yangli Wan, Xiaowen Wang, Liu Yang, Qianhong Li, Xuting Zheng, Tianyi Bai, Xin Wang

**Affiliations:** College of Food Science and Engineering, Northwest A&F University, Xianyang 712100, China

**Keywords:** biofilm, cell membrane integrity, juglone, *Staphylococcus aureus*, wound healing

## Abstract

A serious problem currently facing the field of wound healing is bacterial infection, especially *Staphylococcus aureus* (*S. aureus*) infection. Although the application of antibiotics has achieved good effects, their irregular use has resulted in the emergence of drug-resistant strains. It is thus the purpose of this study to analyze whether the naturally extracted phenolic compound, juglone, can inhibit *S. aureus* in wound infection. The results show that the minimum inhibitory concentration (MIC) of juglone against *S. aureus* was 1000 μg/mL. Juglone inhibited the growth of *S. aureus* by inhibiting membrane integrity and causing protein leakage. At sub-inhibitory concentrations, juglone inhibited biofilm formation, the expression of α-hemolysin, the hemolytic activity, and the production of proteases and lipases of *S. aureus*. When applied to infected wounds in Kunming mice, juglone (50 μL juglone with a concentration of 1000 μg/mL) significantly inhibited the number of *S. aureus* and had a significant inhibitory effect on the expression of inflammatory mediators (TNF-α, IL-6 and IL-1β). Moreover, the juglone-treated group promoted wound healing. At the same time, in animal toxicity experiments, juglone had no obvious toxic effects on the main tissues and organs of mice, indicating that juglone has good biocompatibility and has the potential to be used in the treatment of wounds infected with *S. aureus*.

## 1. Introduction

The optimum growth temperature of *S. aureus* is 37 °C. It exists in the skin, nasal cavity, throat, gastrointestinal and other parts of many healthy people and is the main pathogen of skin or soft tissue infectious diseases [1]. Skin wound infection caused by *S. aureus* is a serious public problem that interferes with the quality of life of many people and has psychological and socioeconomic effects [2]. *S. aureus* is one of the most commonly found bacteria in wound contamination, mainly due to its remarkable ability to adapt to different environments and to acquire genes associated with antimicrobial resistance [3].

A key factor associated with *S. aureus* pathogenicity and drug resistance is the biofilm, an extracellular polymer (EPS) secreted by *S. aureus*. Biofilm can protect bacteria from antibacterial substances, because when bacteria are covered by biofilm, only a small number of drug components, if any, can penetrate it, so the efficacy of drugs is greatly reduced or even lost [4]. In addition, during *S. aureus* infection, α-hemolysin (Hla) is involved in the activation of immune signaling and can directly trigger an inflammatory response by activating recognition receptors that promote inflammatory factors (such as TNF-α, IL-1β, and IL-6) generation [5]. Equally important to the virulence of *S. aureus*, along with biofilms and hemolysins, are proteases and lipases, which can cause cell damage and infection progression [6]. At present, the conventional treatment of *S. aureus* infection is to use antibiotics, which has a certain therapeutic effect. Misuse of antibiotics has accelerated the development and spread of resistance mechanisms, so that many strains now circulating are resistant to certain antibiotics, such as methicillin-resistant *Staphylococcus aureus* (MRSA). Therefore, it is urgent to find novel and natural anti-*S.aureus* substances.

Juglone (5-hydroxy-1,4-naphthoquinone, Figure 1A) is a natural naphthoquinone-structured chemical isolated from the roots, stems, leaves and fruits of the walnut tree [7]. Previous studies have shown that juglone has antibacterial [8], antiviral [9], antifungal [10] and antioxidant effects [7]. However, there are relatively few mechanistic studies on the effect of juglone on *S. aureus*. The aim of this study was to deeply explore the antibacterial mechanism of juglone against *S. aureus*. Furthermore, the topical therapeutic efficacy of juglone was evaluated by means of a mouse model of a skin wound contaminated with *S. aureus*, and the question of whether juglone could reduce the load of *S. aureus* in the infected skin and accelerate the process of wound healing was analyzed. Furthermore, we sought to provide data support for juglone as a novel antibacterial agent for the treatment of *S. aureus*-infected wounds.

## 2. Results

### 2.1. Minimum Inhibitory Concentration (MIC) and Minimum Bactericidal Concentration (MBC) of Juglone against S. aureus

As shown in Table 1, the minimum inhibitory concentration of juglone against three strains (ATCC29213, ATCC43300, ST97) was 1000 μg/mL, and the minimum bactericidal concentration was 2000 μg/mL. The sources of the three strains are shown in Table 1. Due to the clear genetic background of strain ATCC29213 and its strong ability to form biofilms [11], it was used as the research object in subsequent experiments to illustrate the antibacterial mechanism of juglone.

### 2.2. Growth Curve of S. aureus and SIC of Juglone on S. aureus

As shown in Figure 1B, when the concentrations of juglone were 1/2 × MIC, MIC, and 2 × MIC, the growth of *S. aureus* (ATCC29213) was significantly inhibited. When the concentrations of juglone were 1/4 × MIC and 1/8 × MIC, the time for *S. aureus* to reach the stationary phase was delayed, and the OD value in the stationary phase was also relatively low. When the concentrations of juglone were reduced to 1/16 × MIC, 1/32 × MIC, and 1/64 × MIC, there was no significant effect on the growth of the ATCC29213 strain, so 1/16 × MIC, 1/32 × MIC, and 1/64 × MIC were determined to be the SIC of juglone against *S. aureus*.

### 2.3. The Effect of Juglone on the Membrane Potential of S. aureus

It can be seen from Figure 2A that when the *S. aureu*s ATCC29213 strain was treated with the MIC concentration of juglone, the membrane potential was significantly reduced (*p* < 0.01) by 31.10% compared with the control group (treated with the corresponding concentration of DMSO). Similarly, 2 × MIC concentration of juglone also reduced the membrane potential of *S. aureus* to 67.44% (*p* < 0.01). This indicates that juglone significantly reduced the membrane potential of *S. aureus*, resulting in the hyperactivation of *S. aureus* cells.

### 2.4. The Effect of Juglone on the Intracellular ATP Concentration of S. aureus

In order to determine the intracellular ATP concentration of *S. aureus* (ATCC29213), a standard curve was first established using the standard in the ATP detection kit. The obtained standard curve was y = 828802x, and R^2^ = 1, which indicates that the established standard curve was reliable. The ATP concentration of each sample was then calculated according to the standard curve. As shown in Figure 2B, the MIC concentration of juglone significantly decreased the intracellular ATP concentration of *S. aureus* (*p* < 0.01), which was 16.75%. When the concentration of juglone increased to 2 × MIC, the intracellular ATP concentration of *S. aureus* decreased to 12.72%. This indicates that juglone treatment significantly inhibited the intracellular ATP concentration of *S. aureus*, in a dose-dependent manner.

### 2.5. The Effect of Juglone on Total Protein Expression in S. aureus

In order to detect the expression of *S. aureus* (ATCC29213) intracellular protein, the extracted total protein was subjected to SDS-PAGE gel electrophoresis and then stained with Coomassie brilliant blue. As shown in Figure 2C,D, the control group had bright protein bands, while treatment of *S. aureus* with juglone reduced the abundance of intracellular proteins in *S. aureus* (*p* < 0.01), regardless of whether the concentration of juglone was 1/2 × MIC or MIC, or 2 × MIC.

### 2.6. The Effect of Juglone on the Cell Membrane Integrity of S. aureus

In order to evaluate the effect of juglone on the membrane integrity of *S. aureus* (ATCC29213), a standard curve of cell viability was first established according to the instructions of the Live/Dead^®^Baclight TM kit. The established standard curve was y = 0.0113x + 0.059, and R^2^ = 0.997, indicating that the established standard curve has good reliability. Cell viability was then calculated for each sample based on the established standard curve. As shown in Figure 2E, the proportion of bacteria with intact cell membranes decreased by 85.39% after treatment with juglone at the MIC concentration. When the juglone concentration increased to 2 × MIC, the proportion of bacteria with intact cell membrane decreased to 2.54%. This indicates that juglone significantly damaged the membrane integrity of *S. aureus* in a dose-dependent manner. Consistent with expectations, the laser confocal results also showed (Figure 3) that, when *S. aureus* was treated with MIC concentration of juglone, the number of cells stained with PI increased significantly, and with the increase of juglone concentration to 2 × MIC, the red fluorescence intensity was enhanced, indicating an increased proportion of membrane damaged cells.

### 2.7. The Effect of Juglone on the Cell Morphology of S. aureus

As shown in Figure 4, the surface of *S. aureus* in the control group was smooth and round, and evenly arranged like a grape bunch. When *S. aureus* was treated with juglone, the cells shrank, and the intracellular contents were leaked out. And with the increase of juglone concentration, shrinkage and spillage of contents were exacerbated.

### 2.8. Inhibition of S. aureus Biofilm Formation by Juglone

In order to analyze the effect of juglone on the biofilm formation of *S. aureus*, crystal violet staining was performed after *S. aureus* (ATCC29213) was treated with a sub-inhibitory concentration of juglone for 24 h. When the concentration of juglone was 1/16 × MIC, the formation of *S. aureus* biofilm was significantly inhibited (*p* < 0.01), which can be seen from Figure 5. Moreover, the inhibitory effect on *S. aureus* biofilm formation was weakened when the concentration of juglone was reduced to 1/32 × MIC. This indicates that juglone had an inhibitory effect on the biofilm formation of *S. aureus* in a dose-dependent manner.

### 2.9. Inhibition of Hemolytic Activity of S. aureus and Expression of Toxins (Hla) by Juglone

Hemolytic ability is an important pathogenic factor of *S. aureus*, so in order to evaluate whether juglone has an effect on the hemolytic activity of *S. aureu* (ATCC29213) *s*, 6% of the fresh sheep red blood cells were used for hemolysis analysis. As shown in Figure 6A,B, juglone significantly inhibited the hemolytic activity of *S. aureus* (*p* < 0.01), which decreased the hemolytic activity to below 15.46%, with an inhibition that was dose-dependent. Consistent with the hemolytic activity, Western blot results (Figure 6C,D) also show that sub-inhibitory concentrations of juglone also significantly inhibited the expression of *S. aureus* protein Hla (*p* < 0.01).

### 2.10. Juglone Inhibited the Expression of Protease and Lipase of S. aureus

As shown in Figure 7, after *S. aureus* (ATCC29213) was treated with sub-inhibitory concentrations of juglone for 24 h, the expression of protease and lipase were inhibited. Additionally, with the increase of juglone concentration, the inhibitory effect gradually strengthened.

### 2.11. Topical Juglone Treatment Accelerates the Shrinkage of Wounds Infected with S. aureus

Next, we evaluated the efficacy of topical treatment with juglone in a model of *S. aureus* (ATCC29213)-induced excisional wound infection. As expected, macroscopic analysis of lesions showed that juglone treatment promoted faster healing and earlier granulation tissue formation compared with the untreated group, achieving similar results as the vancomycin-treated group (Figure 8).

### 2.12. Topical Treatment with Juglone Reduces Bacterial Burden in S. aureus-Infected Wounds and Improves Histological Evaluation

Bacterial load in the wound tissue was quantified on day two of the treatment. As shown in Figure 9A, both the vancomycin treatment group and the juglone treatment group significantly reduced the load of *S. aureus* (ATCC29213) in the wound compared with the PBS treatment group (*p* < 0.01).

According to histological analysis, the healing process was accelerated in the juglone and vancomycin treatment group, and there was obvious epithelial regeneration after 14 d of treatment. Compared with the *S. aureus* infection group, the connective tissue of the dermis of the juglone-treated mice became denser, the collagen fiber bundles were evenly distributed, and the number of sweat glands in the subcutaneous tissue increased (Figure 9B), which was consistent with the state of the vancomycin treatment group.

### 2.13. Topical Treatment of Juglone Modulated the Levels of Inflammatory Factors of Mice e with S. aureus-Infected Wounds

Subsequently, to analyze whether the promotion of wound healing by juglone was related to immune regulation, we measured cytokine levels in serum and in wound tissue on day three after *S. aureus* (ATCC29213) infection. As shown in Figure 10, the serum levels of TNF-α, IL-6 and IL-1β in mice were significantly increased after *S. aureus* infection (*p* < 0.01), whether in serum or wound tissue. Additionally, compared with the control group, juglone treatment had an inhibitory effect on the expression of three inflammatory factors, but the difference was not significant (*p* > 0.05). Notably, topical treatment of *S. aureus*-infected wounds with juglone significantly reduced the expression of inflammatory factors of mice compared with the untreated group (*p* < 0.05 or *p* < 0.01), whether in serum or wound tissue.

### 2.14. Biosafety Evaluation of Juglone

To evaluate the safety of juglone, the wounds of mice were treated with juglone continuously for 14 d, and the skin was stained with H&E. As shown in Figure 11, compared with the control group (treated with PBS), there was no obvious damage to the major organs of mice by juglone, indicating that juglone has better biocompatibility.

## 3. Discussion

*S. aureus* infection of skin wounds is a public health hazard due to its increasing incidence. The usual treatment option is the use of antibiotics, which does have some effect, but chronic antibiotic abuse has led to the emergence of resistant strains [12]. However, the rate of discovery and development of novel antibiotics is nowhere near enough to solve the problem. Since *S. aureus*-infected wounds cause a significant inflammatory response, in this study we analyzed the antibacterial activity of juglone with anti-inflammatory effect on *S. aureus* and explored its mechanism. At the same time, with the help of the wound model of *S. aureus* infection in mice, it was confirmed that juglone has the ability to promote the wound healing of skin wounds infected with *S. aureus*. In terms of strain selection, ATCC29213 (MSSA) and ATCC43300 (MRSA) were purchased from the American type culture collection as these are the standard model strains of *S. aureus,* with well-established genetic information. ST97 (MRSA) was obtained from cows with clinical mastitis and reacts to clinical significance.

The active substance juglone in this study is a class of naphthoquinones, which are used as medicinal raw materials in many countries due to their antibacterial, anti-inflammatory, antiviral and antioxidant effects [13]. Compared with other natural products with the same antibacterial effect, juglone is mainly extracted from the waste of walnut processing—walnut peel [14]. The raw materials are easy to obtain and are cheap, the extraction process is mature, and it has a stable source of supply. Today, juglone is also found in colorant applications in the food, cosmetic and textile industries [15]. One study found that the MIC value of walnut juglone and erythromycin against another standard strain of *S. aureus*—25,923—was consistent. Studies have shown that the MIC of novel juglone and naphthalene derivatives against methicillin-resistant *S. aureus* is 8 μg/mL [16]. The results of this study are consistent with the above results, confirming that juglone has antibacterial activity against *S. aureus*. This antibacterial activity may be related to the naphthoquinone structure of juglone, as our previous study showed that shikonin has good anti-*S. aureus* activity because it has a naphthoquinone structure [17]. As another example, plumbagin has better antibacterial activity against methicillin-resistant *Staphylococcus aureus* [18].

As a key factor for bacterial growth, the bacterial cell membrane not only plays a key role in physiological metabolism, but also acts as a barrier to protect bacteria from foreign harmful substances [19]. Cell membrane integrity can be assessed by analyzing cell morphology, cell membrane potential, cell viability, and leakage of intracellular protein and ATP components.

Zw A et al. reported that xanthan gum oligosaccharide (LW-XG) achieved a bacteriostatic effect on *S. aureus* by damaging the cell membrane rather than the cell wall [20]. Ibtihel et al. demonstrated that the antibacterial properties of polysaccharides extracted from olive leaves may be due to disruption of bacterial cell walls and membranes [21]. Studies by Luther et al. show that antimicrobial peptides inhibit the growth of pathogenic bacteria by disrupting cell walls or membranes [22]. Through scanning electron microscopy and transmission electron microscopy observations, Shu et al. found that bacterial cells treated with mannose erythritol lipid MELs exhibited vacancies at the edge of the cell wall, cytoplasmic shrinkage and plasmid division, further confirming that the integrity of the cell membrane was damaged [23]. The electron microscope results of this study were similar to the above studies, and the juglone treatment caused the *S. aureus* cells to shrink and the cell membrane to sink. This suggests that in the future development of antibacterial substances, the cell membrane of bacteria can be used as an important target.

Membrane potential (MP) represents the potential difference between the inside and outside of bacteria and plays a crucial role in bacterial metabolism. The negative charge anionic fluorescent dye DiBAC_4_(3) combines with the protein in the cytoplasm to emit fluorescence, thus reflecting the negative external and positive internal state on both sides of the membrane, which is called its polarization state. At this time, the Na^+^ and K^+^ voltage gated system is closed. When the membrane potential changed in the direction of increasing negative value in the membrane, K^+^ continued to flow out of the cell, and the decrease of intracellular fluorescence intensity indicates that the cell had hyperpolarization. On the contrary, when the membrane potential changes in the direction of a decrease in the negative value in the membrane, the Na^+^ channel opens and the intracellular Na^+^ concentration increases. The increase of intracellular fluorescence intensity indicates the depolarization state of cells. The decrease in MP may be due to structural damage to the cell membrane [24]. Studies have shown that the fluorescence intensity of cell membrane decreased after ginger essential oil (GEO) treatment of bacterial cells, indicating that hyperpolarization of cell membrane leads to a decrease in cell metabolic activity. The study by Lu et al. showed that the sorafenib derivative, SC5005, shows bactericidal activity against MRSA in vivo and in vitro by dissipating membrane potential and does not induce resistance in the strain [25]. Consistent with our findings, juglone significantly reduced the membrane potential of *S. aureus*, resulting in the hypertrophy of bacterial cells, which in turn damaged the integrity of the cell membrane.

Bacterial viability refers to the ability of cells to grow and reproduce under a defined set of environmental conditions. Today, cell viability defines the state in which cells are able to perform various aspects of metabolic, physiological and genetic functions, as well as the degree of morphological integrity [26]. Propidium iodide (PI) is a basic type that can only bind to the DNA of cells with damaged cell membranes and displays red fluorescence. SYTO 9 can label all bacterial DNA in the population and display green fluorescence. When both were added to the damaged cell membrane phage system at the same time, PI caused a decrease in the staining intensity of SYTO 9. The more severely damaged the cell membrane, the stronger the red fluorescence intensity. In this study, when *S. aureus* was treated with MIC concentration of juglone, almost all the cells showed red fluorescence, which indicated that the treatment of juglone damaged the integrity of the *S. aureus* cell membrane and decreased the cell viability.

Protein is the material basis and main carrier of life activities, and is closely related to various life forms [27]. Studies have shown that ginger essential oil exerts its antibacterial activity by destroying the integrity of cell membranes and causing protein leakage in bacteria [28]. The results of this study show that juglone-treated *S. aureus* significantly reduced the protein synthesis of bacterial cells and exerted a bacteriostatic effect. Additionally, the sub-inhibitory concentration of juglone significantly inhibited the secretion of protease and lipase of *S. aureus*, which may be one of the mechanisms by which juglone has anti- *S. aureus* activity.

Stable intracellular ATP concentration is necessary to maintain the normal life activities of *S. aureus*. The study of Zw A et al. showed that LW-XG inhibited the activity of Ca^2+^-Mg^2+^-ATPase on the cell membrane, thereby affecting the concentration of intracellular ATP in *S. aureus* and finally producing a bacteriostatic effect [20]. ATP synthase has been validated as a druggable target, and studies have shown that inhibition of ATP synthase is a strategy to eliminate pathogens such as *S. aureus* [29]. Our results also show that juglone treatment caused a decrease in the intracellular ATP concentration of *S. aureus*, which may be due to the leakage of ATP caused by the damage to the cell membrane integrity, or it may be that juglone inhibited the activity of ATPase, questions which need to be further explored in the future.

Biofilms provide a relatively stable living environment for the survival, proliferation, and recolonization of bacteria, so they can be used as important targets for drugs. Xanthan gum oligosaccharide (LW-XG) inhibits the formation of *S. aureus* biofilm by inhibiting the transcription of genes fnbA, fnbB and clfB, thereby exerting antibacterial activity [20]. The inhibitory and degrading effects of grapefruit seed extract (GSE) on *S. aureus* and Escherichia coli biofilms have also been observed by crystal violet staining method [30]. Bai et al. have also reported that shikimic acid significantly reduces the biomass of biofilm in SICs, inhibits the adhesion of biofilm and destroys the structure of *S. aureus* biofilm [11]. 20S-Ginsenoside Rg3 (Rg3) inhibits the formation of *S. aureus* biofilm by inhibiting the SaeR/SaeS two-component system [31]. In addition, quorum sensing regulates biofilm formation by regulating the synthesis of extracellular polysaccharide and intercellular polysaccharide adhesion. Yang et al. (2019) have shown that coenzyme Q_0_ inhibits biofilm formation of Salmonella typhimurium by interfering with quorum sensing [32]. Consistent with the above studies, our findings also show that juglone treatment significantly inhibited biofilm formation of *S. aureus*. However, whether juglone can inhibit the virulence of *S. aureus* by a quorum sensing system will be our next research direction.

Hla is one of the main factors leading to the pathogenicity of *S. aureus*, which manifests as a secreted 33.3 kDa water-soluble monomer, and which in turn acts after oligomerization into heptamers on the host cell membrane. Many antibacterial substances exert their antibacterial activity by inhibiting the expression of Hla. For example, lycopene reduces inflammation caused by *S. aureus* by inhibiting the expression of Hla [33]. Our results also show that juglone had a significant inhibitory effect on the expression of Hla in *S. aureus*, thereby exhibiting an inhibitory effect on the hemolytic activity of *S. aureus*.

*S. aureus* infection is an important factor causing delayed wound healing. Our findings suggest that topical treatment of juglone promotes the healing of *S. aureus*-infected wounds, which might be due to the juglone-related inhibition of *S. aureus* load in the wound, or juglone’s anti-inflammatory effects, which inhibit the levels of inflammatory factors (TNF-α, IL-6 and IL-1β). TNF-α is strongly associated with the severity of skin infections caused by *S. aureus* [34]. Studies have shown that Cratylia mollis lectin significantly improved the severity of *S. aureus* infected lesions by inhibiting the expression of TNF-α in the lesions [35]. Similarly, in this study we demonstrated that topical treatment of juglone also significantly inhibited the level of IL-6 and IL-1β in mice, which is consistent with previous studies showing that juglone has anti-inflammatory effects [7].

## 4. Materials and Methods

### 4.1. Strains and Culture

*S. aureus* ATCC29213 (methicillin-sensitive *Staphylococcus aureus*, MSSA), ST97 (MRSA), ATCC43300 (MRSA) strains were stocked by our laboratory. Strains were first streaked on TSA plates for activation. After 24 h incubation at 37 °C, a single colony was picked and inoculated into 5 mL of TSB and incubated at 37 °C for 16 h at 180 rpm. The bacterial solution was measured for optical density (OD) at 600 nm with a spectrophotometer, the OD value was adjusted to 0.5, and the bacterial concentration at this time was about 10^8^ CFU/mL. Then, a 100-fold dilution was performed to obtain a concentration of about 10^6^ CFU/mL bacterial suspension for subsequent experiments.

### 4.2. MIC and MBC

The determination of the MIC of juglone (Shanghai Yuanye Biotechnology Co., Ltd., Shanghai, China) against *S. aureus* referred to the guidelines of the Clinical and Laboratory Standards Institute, with minor modifications [36]. Juglone was added to 200 μL of bacterial suspension (approximately 10^6^ CFU/mL), so that the final concentrations were 4000 μg/mL, 2000 μg/mL, 1000 μg/mL, 500 μg/mL, 250 μg/mL, 125 μg/mL, 62.5 μg/mL, 31.25 μg/mL, 15.625 μg/mL, 7.8125 μg/mL, and 0 μg/mL (control group). At the same time, 32 μg/mL of vancomycin was added to the bacterial suspension as a positive control group. The corresponding concentrations of juglone bacterial solution without *S. aureus* were used as blank control groups. After 24-h incubation at 37 °C, the OD value was determined at 600 nm with a multifunctional microplate reader (spark, Austria). The minimum juglone concentration with the difference in OD value (experimental group-blank control group) of less than 0.05 was defined as the MIC. The above experimental group culture solution was spread on a TSA plate, and after culturing at 37 °C for 24 h, the minimum juglone concentration without colony growth was defined as the MBC.

### 4.3. Growth Curve and Subinhibitory Concentration (SIC)

Growth curves were determined as previously described [17]. Juglone was dissolved in bacterial suspension to give final concentrations of 0 (control group), 1/128 × MIC, 1/64 × MIC, 1/32 × MIC, 1/16 × MIC, 1/8 × MIC, 1/4 × MIC, 1/2 × MIC, MIC, and 2 × MIC. An amount of 200 μL of the above mixture was added to a honeycomb plate, which was then placed in an automatic microbial growth curve analyzer (Labsystems, Helsinki, Finland), the temperature was set to 37 °C, and the OD value (with a wavelength of 600 nm) was measured every hour for 24 h continuously. Growth curves were drawn based on the measured OD values. The juglone concentration that had no significant effect on *S. aureus* was defined as the SIC.

### 4.4. Cell Membrane Potential

The determination of *S. aureus* (ATCC29213) cell membrane potential was based on the method of Guo et al. [37], with some minor modifications. An amount of 200 μL of the bacterial suspension was added to a 96-well black microtiter plate, followed by addition of juglone to final concentrations of 2 × MIC, MIC, and 0 (the corresponding volume of DMSO was added as a control), respectively. The microtiter plate was placed at 37 °C, and after 1 h of incubation, DIBAC_4_(3) (Sigma-Aldrich, St. Louis, MO, USA) was added to each well so that its final concentration was 1 μM. After a 10-min incubation in the dark, the fluorescence intensity of all samples were measured on a microplate reader (spark, Austria) (excitation wavelength was 492 nm, emission wavelength was 515 nm). The fluorescence values measured by different concentrations of juglone solution (without bacterial suspension) were used as the background fluorescence values. The actual fluorescence values of different experimental treatment groups was the difference between the measured value and the corresponding background fluorescence value.

### 4.5. Intracellular ATP Concentration

The ATP concentration was determined as described in previous experiments [17]. An amount of 2 mL of *S. aureus* (ATCC29213) bacterial suspension was treated with three concentrations of juglone (2 × MIC, MIC, 0) for 1 h, then centrifuged at 6000 rpm for 10 min, and the bacterial pellet was collected. The cells were then washed three times with PBS and finally resuspended in 200 μL of lysis buffer. The samples were placed in an ultrasonic apparatus for 10 min (power 25 W, sonicated for 6 s, and stopped for 3 s). The ultrasound was completed when it was observed that the liquid had become transparent. After centrifugation at 12,000 rpm for 10 min at 4 °C, the supernatant was collected for subsequent experiments. The determination of intracellular ATP concentration was performed according to the instructions of the ATP detection kit (Beyotime, Shanghai, China).

### 4.6. Intracellular Proteins

The determination of bacterial intracellular proteins was performed as described by Han et al. [38], but with some minor modifications. An amount of 4 mL of *S. aureus* (ATCC29213) bacterial suspension was treated with juglone at final concentrations of 2 × MIC, MIC, 1/2 × MIC and 0 (control group) for 1 h. The cells were collected by centrifugation at 6000 rpm for 10 min. Cells were then washed three times with PBS and finally resuspended in 400 μL of lysis buffer, at which time protease inhibitor (PMSF) was also added to prevent protein degradation during sonication. The samples were then placed in a sonicator and sonicated on ice for 10 min. After centrifugation at 12,000 rpm for 10 min, the supernatant was collected to obtain whole bacterial protein for subsequent experiments. The protein loading buffer was added to the whole bacterial protein samples obtained above. After mixing, the samples were placed in boiling water for thermal denaturation for 5 min, and then stored at −20 °C. Five percent stacking gel and 12% resolving gel were prepared for SDS-PAGE protein gel electrophoresis. An amount of 10 μL of protein samples were loaded into the gel wells, electrophoresed at 70 V for 40 min, and then electrophoresed at 120 V for 70 min. After the gel was stained in Coomassie Brilliant Blue for 1 h, destaining solution was added and destained overnight.

### 4.7. Cell Membrane Integrity

The determination of cell membrane integrity was followed the research of Kang et al. [39]. Various doses of Juglone was added into *S. aureus* (ATCC29213) bacterial suspensions at concentrations of 2 × MIC, MIC, 0 (control group). After 1 h, the cells were collected, washed three times with 0.9% NaCl, and finally resuspended with 1 mL 0.9% NaCl. An amount of 3 μL of propidium iodide (PI) and SYTO 9 mixed dye (1:1 mix) was added to each sample. After mixing, we let the samples stand in the dark for 10 min. An amount of 200 μL of the mixture was added to a black microtiter plate (Thermo Fisher, Waltham, MA, USA) and placed in a multi-plate reader (spark, Austria) to measure the fluorescence intensities of PI (excitation wavelength was 485 nm, emission wavelength was 610 nm) and SYTO 9 (excitation wavelength was 485 nm, emission wavelength was 542 nm), respectively. The cell viability standard curve was established according to the instructions of the Live/Dead^®^BaclightTM kit (Sigma-Aldrich, St. Louis, MO, USA), and then the cell viability of different samples was calculated.

### 4.8. Confocal Laser Scanning Microscopy (CLSM)

As described in Section 2.7, the samples treated with different concentrations of juglone were mixed with PI and SYTO 9 fluorescent dyes and incubated in the dark for 10 min. An amount of 20 μL of the mixture was pipetted and placed on a glass slide, which was finally observed under a laser confocal microscope (Leica, TCS SP8, Osaka, Japan) and photographed for recording.

### 4.9. Field Emission Scanning Electron Microscopy (FE-SEM)

Field emission scanning electron microscopy was based on the method of Shi et al. with some modifications [40]. First, 1 mL of *S. aureus* (ATCC29213) suspension was treated with juglone (final concentrations of 2 × MIC, MIC, 0) for 1 h, centrifuged at 6000 rpm for 10 min, and the cells were harvested. The cells were washed three times with PBS and fixed with 1 mL of 2.5% glutaraldehyde overnight at 4 °C. The cells were then dehydrated with alcohol in a concentration gradient (30%, 50%, 70%, 80%, 90%, 100%) for 20 min, and then the samples were mixed with tert-butanol and dried. Finally, the samples were sprayed with gold, and observed and photographed under a scanning electron microscope (Tokyo, S-4800, Osaka, Japan).

### 4.10. S. aureus Biofilm Formation

The determination of biofilm formation followed the method of Li et al. with some minor modifications [41]. Various doses of juglone were added into different *S. aureus* (ATCC29213) bacterial suspensions at final concentrations of 1/16 × MIC, 1/32 × MIC, 1/64 × MIC, 0 (control). Different concentrations of juglone solutions without bacterial cells were used as blank control groups. The mixture was then added to a 96-well polystyrene plate, incubated at 37 °C for 24 h, and the optical density of each well was measured at 600 nm. The cultures were gently discarded, and each well was gently rinsed three times with PBS, fixed with absolute ethanol for 10 min, and placed in an oven at 65 °C for 20 min. Then, 1% crystal violet was added to stain for 15 min before being gently discarded and rinsed three times with PBS again to remove unbound crystal violet. After drying the 96-well plate in an oven for 20 min, 33% glacial acetic acid was added to dissolve the biofilm. The optical density of the sample wells was then determined at 570 nm. Finally, the generated amount of biofilm was calculated according to Formula (1):(1)$$\mathrm{SBF}=\frac{OD570(\mathrm{treat})-OD570(\mathrm{blank})}{OD600(\mathrm{treat})-OD600(\mathrm{blank})}$$

### 4.11. Hemolytic Activity

The determination of hemolytic activity of *S. aureus* (ATCC29213) was undertaken in reference to the study of Teng et al. [42], with slight modifications. Juglone was added to 1 mL of bacterial suspension to a final concentration of 1/16 × MIC, 1/32 × MIC, 1/64 × MIC, 0 (control). The mixture was incubated at 37 °C for 24 h, centrifuged at 10,000 rpm for 10 min, and the supernatant was harvested. An amount of 250 μL of culture supernatant was added to 800 μL of 6% fresh sheep red blood cells, placed in a 37 °C incubator for 30 min, and centrifuged at 6000 rpm, 4 °C for 10 min. The supernatant was gently pipetted and placed at 450 nm to measure the optical density. A sample to which 250 μL of 1% Triition X-100 was added was used as a positive control, defined as 100% hemolysis. The ratio of sample to positive control is the percent hemolysis.

### 4.12. Hla Expression

As described in Section 2.11, after the culture supernatant was harvested, 2 X protein loading buffer was added at a ratio of 1:1, the protein was denatured in boiling water for 5 min, and then loaded onto a protein gel (5% stacking gel, 12% resolving gel) for electrophoresis. After electrophoresis, it was transferred to PVDF membrane. Membranes were blocked in 5% nonfat dry milk for 4 h, and then incubated with Hla (Abcam, Cat. No. ab52922) primary antibodies overnight at 4 °C. After five washes, the membrane was incubated with secondary antibody for 1 h at room temperature. Six rinses were performed again. Finally, it was placed under a chemiluminescence imager (Tanon, Shanghai, China) to develop and photograph.

### 4.13. Protease Production Assay

First, *S. aureus* (ATCC29213) was inoculated on TSA plates containing 3% nonfat dry milk and cultured at 37 °C for 24 h. When a transparent circle appears around the colony, this indicates that the bacteria has the ability to produce protease. Subsequently, the supernatant was harvested as described in Section 2.11. An amount of 500 μL of the supernatant was added to 1 mL of 1.25% skim milk, incubated at 37 °C for 1 h, and then placed at 600 nm to measure the optical density. TSB was added to skim milk as a blank control. The percentage of protease production was in reference to Formula (2):(2)$$Protease\;production\left(\%\right)=\frac{OD600\left(blank\right)-OD600\left(treat\right)}{OD600\left(blank\right)-OD600\left(Control\right)}\times 100\%$$

### 4.14. Lipase Production Assay

The strains were first inoculated on TSA plates containing 10% egg yolk emulsion and incubated at 37 °C for 48 h. When a transparent circle appears around the colony, this indicates that the colony has the ability to produce lipase. After *S. aureus* (ATCC29213) was treated with sub-inhibitory concentrations of juglone, the supernatant was harvested as described in Section 2.11. An amount of 500 μL of the supernatant was added to 1 mL of 10% egg yolk emulsion, incubated at 37 °C for 1 h, and then placed at 600 nm to measure the optical density. TSB was added to egg yolk emulsion as blank control. The percentage of production of lipase refers to Formula (3):(3)$$Lipase\;production\left(\%\right)=\frac{OD600\left(blank\right)-OD600\left(treat\right)}{OD600\left(blank\right)-OD600\left(Control\right)}\times 100\%$$

### 4.15. Mouse Wound Model and S. aureus Infection

Animal experiments were approved by the Animal Protection and Utilization Committee of Northwest Agriculture and Forestry University and were conducted following the guidelines of the Animal Protection and National Institutes of Health [35]. Twenty-five healthy Kunming mice (six to eight weeks old) were housed in a ventilated room with an average temperature of 21 °C and a light cycle of 12 h, with free access to food and water. After one week of adaptive feeding, the mice were randomly divided into five groups. For experimental surgical procedures, mice were pre-anaesthetized with 1 mg/kg xylazine hydrochloride and 50 mg/kg ketamine intramuscularly. After each mouse was anesthetized, the hair in the thoracic and dorsal region was shaved, and after sterilizing with alcohol wipes, a wound of 8 mm in diameter was formed by cutting the skin with sterile scissors. The wound of three groups of mice was infected with 100 μL of *S. aureus* (ATCC29213) suspension (10^7^ CFU/mL). One day after infection, a group of mice (*n* = 5) that received juglone (MIC dose) treatment was designated the *S. aureus* + juglone group. A group of mice (*n* = 5) that received vancomycin treatment was designated the *S. aureus* + vancomycin group. Another group of mice was treated with PBS every day, and this became the *S. aureus* group. At the same time, a group of mice that were not inoculated with *S. aureus* in the wound area were treated with PBS only every day, forming the control group (*n* = 5); the mice treated with only juglone formed the juglone group (*n* = 5). The whole treatment process was carried out in a well-ventilated experimental environment. The wound was covered with medical gauze after each treatment to prevent contamination by other micro-organisms. Mice were observed and monitored daily for wound healing. On the 14th day, the mice were euthanized, and their skins were collected for hematoxylin eosin (H&E) staining.

### 4.16. Biocompatibility Analysis of Juglone

In order to analyze the biocompatibility of juglone, the main organs (heart, liver, spleen, lung, and kidney) of the mice were obtained for H&E staining after being treated with juglone for 14 days [43].

### 4.17. Macroscopic Assessment of Lesions

As described in Section 4.15, on days 0, 3, 7, and 14 of treatment, the wounds were photographed and recorded, and the wound size was calculated using Image J (National Institutes of Health, 1.43.67, Bethesda, MD, USA). The degree of wound healing is expressed by Equation (4):(4)$$Wound\;healing(\%)=\frac{W_{0}-W_{d}}{W_{0}}\times 100\%$$

Among these, *W*_0_ was the initial wound area, and *W_d_* was the measurement day area.

### 4.18. Histological Evaluation

The histological assessment method was as described by Suarez Carneiro et al. with minor modifications [35]. Skin fragments (covering experimentally induced lesion areas and intact skin) were collected at 14 dpi (14 d after the wound was inoculated with *S. aureus* (ATCC29213). Each sample was fixed in 4% paraformaldehyde (pH 7.2). Tissue sections were HE stained. The lesions were analyzed by light microscopy at 200× magnification. Criteria assessed included cellular debris, inflammatory infiltration, re-epithelialization, vascularization, and the distribution pattern of collagen fibers.

### 4.19. Bacterial Loads of Wound Tissue

We referred to the method described by Suarez Carneiro et al. [35]. Wound skin tissue was collected at 2 dpi and placed in 1 mL of PBS, vortexed for five cycles (30 s each), and then centrifuged at 3000 rpm for 5 min. After the supernatant was collected, serial 10-fold dilutions were performed, and appropriate dilution gradients were spread on TSA plates. Bacterial loads were determined and recorded after 24 h of incubation.

### 4.20. Cytokines in Wound Tissue

Cytokines (IL-1β, IL-6, TNF-α) in the serum and wound tissues of each mouse were assayed on day 3 using ELISA kits (Xinlebiology, Shanghai, China).

### 4.21. Statistical Analysis

SPSS version 20.0 (SPSS Inc., Chicago, IL, USA) software was used for data quantification and analysis. Origin 2019 software was used for mapping. All results are presented as mean ± standard deviation. Difference analysis between different groups was performed by analysis of variance (ANOVA), when *p* < 0.05, the difference was considered significant, indicated by *; when *p* < 0.01, the difference was considered extremely significant, indicated by **.

## 5. Conclusions

Overall, juglone showed antibacterial activity against *S. aureus* with an MIC of 1000 μg/mL and an MBC of 2000 μg/mL. The experimental results show that the antibacterial activity was achieved by damaging the cell membrane of the bacteria and increasing the permeability of the cell membrane. Additionally, at sub-inhibitory concentrations, juglone reduced the virulence of *S. aureus* by targeting the formation of bacterial biofilm and the expression of virulence factors (Hla). More importantly, in a mouse model of *S. aureus*-infected skin wounds, we found that juglone with better biocompatibility could accelerate wound healing by inhibiting the proliferation of *S. aureus* in the wound. The above results may provide data support for the potential application of juglone in the pharmaceutical industry.

## Figures and Tables

**Figure 1 ijms-24-03931-f001:**
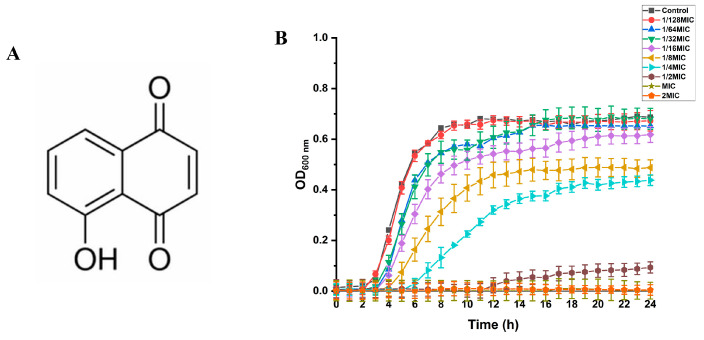
(**A**) Molecular structural formula of juglone. (**B**) The effect of juglone on the growth curve of *S. aureus*.

**Figure 2 ijms-24-03931-f002:**
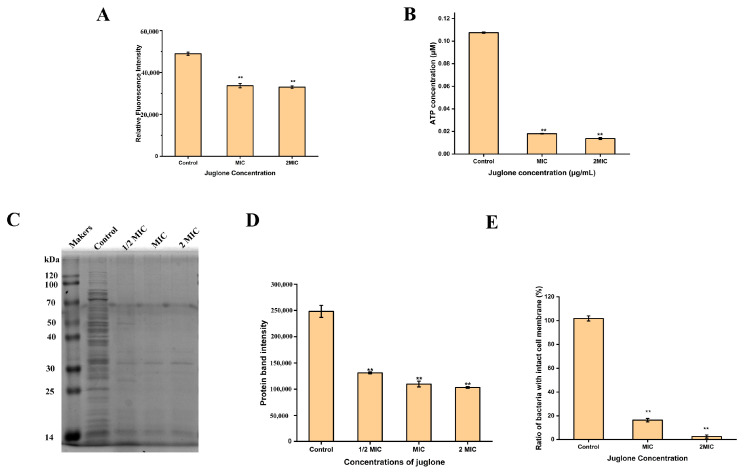
Effect of juglone on intracellular substances of *S. aureus*. (**A**) The effect on the membrane potential of *S. aureus* in the presence of juglone. (**B**) The effect of juglone treatment on the intracellular ATP concentration of *S. aureus*. (**C**,**D**) Detection of *S. aureus* intracellular whole protein after juglone treatment. (**E**) Effects of juglone treatment on *S. aureus* membrane integrity. MIC is 1000 μg/mL. Results are shown as the mean ± standard deviation of three independent experiments. When *p* < 0.01, it is represented by **.

**Figure 3 ijms-24-03931-f003:**
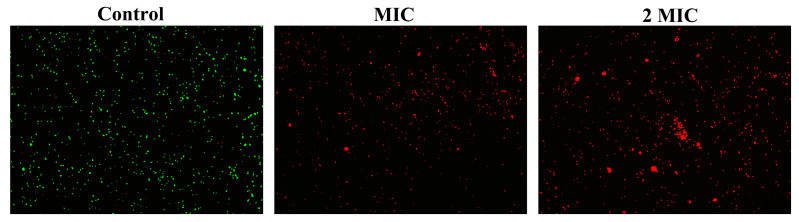
Laser confocal observation of the effect of juglone on the membrane integrity of *S. aureus*. The PI dye label shows red fluorescence and the SYTO 9 dye label shows green fluorescence. The magnification was 200×.

**Figure 4 ijms-24-03931-f004:**
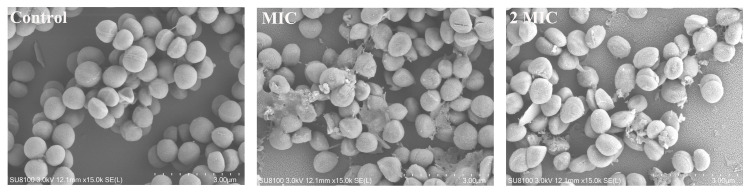
Effect of juglone on the cell morphology of *S. aureus* observed by field emission scanning electron microscopy. The scale bar is 3 μm.

**Figure 5 ijms-24-03931-f005:**
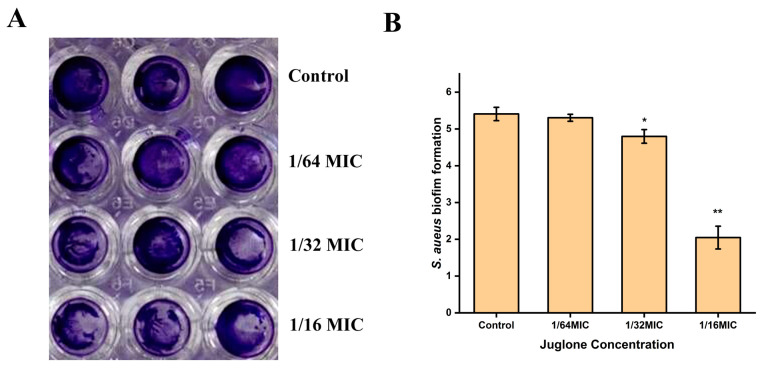
The effect of juglone on *S. aureus* biofilm formation. (**A**) After treatment of *S. aureus* with sub-inhibitory concentration of juglone for 24 h, the formation of biofilm was observed by crystal violet staining. (**B**) Statistical analysis of the effects of sub-inhibitory concentrations of juglone on *S. aureus* biofilm formation. MIC is 1000 μg/mL. When *p* < 0.01, it is represented by **, when *p* < 0.05, it is represented by *.

**Figure 6 ijms-24-03931-f006:**
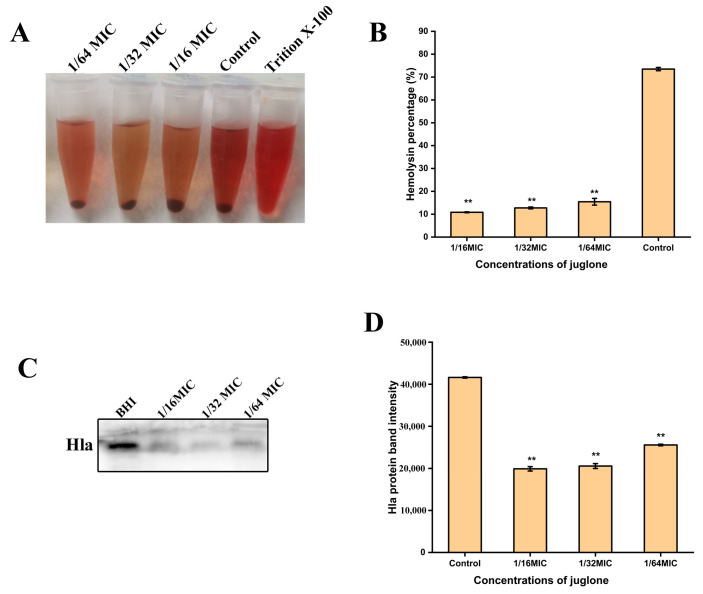
The effect of juglone on the hemolytic activity of *S. aureus*. (**A**,**B**) Six percent fresh sheep erythrocytes were used to analyze the hemolytic activity of *S. aureus* when sub-inhibitory concentrations of juglone were treated with *S. aureus* for 24 h; (**C**,**D**) Western blot analysis of the effect of sub-inhibitory concentration of juglone on protein Hla expression of *S. aureus*. MIC is 1000 μg/mL. When *p* < 0.01, it is represented by **.

**Figure 7 ijms-24-03931-f007:**
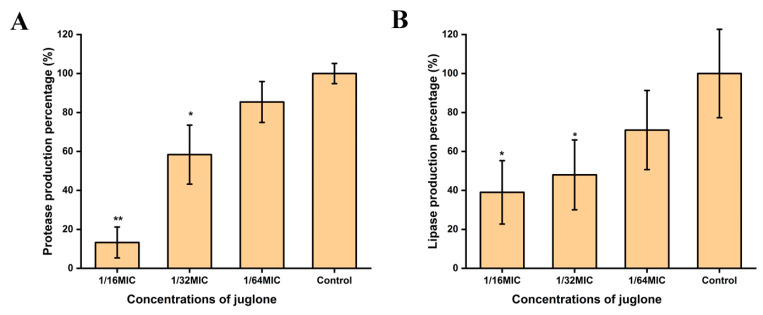
Juglone inhibited the expression of protease and lipase in *S. aureus*. (**A**) The effect of juglone on the protease secretion of *S. aureus*; (**B**) the effect of juglone on the lipase secretion of *S. aureus*. MIC is 1000 μg/mL. When *p* < 0.01, it is represented by **, when *p* < 0.05, it is represented by *.

**Figure 8 ijms-24-03931-f008:**
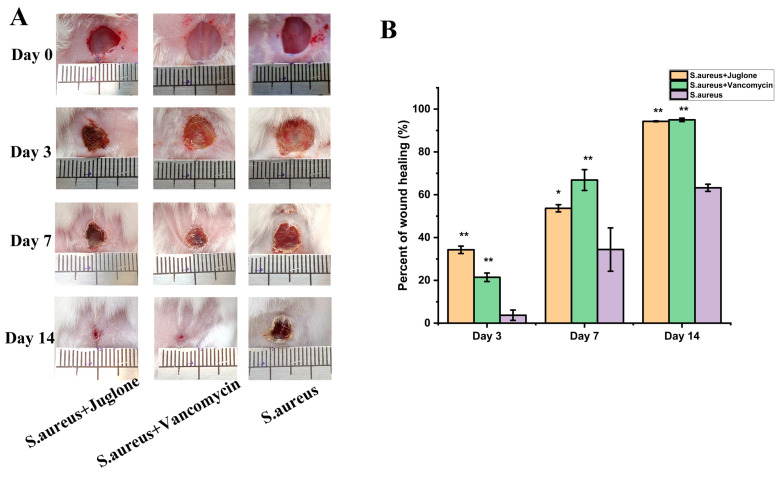
Effect of juglone on wound healing in *S. aureus* infection. (**A**) Representative pictures of *S. aureus* infected wounds on days 3, 7 and 14 of topical treatment with juglone; (**B**) Image J calculation and analysis of the relative wound healing percentage. When *p* < 0.01, it is represented by **, when *p* < 0.05, it is represented by *.

**Figure 9 ijms-24-03931-f009:**
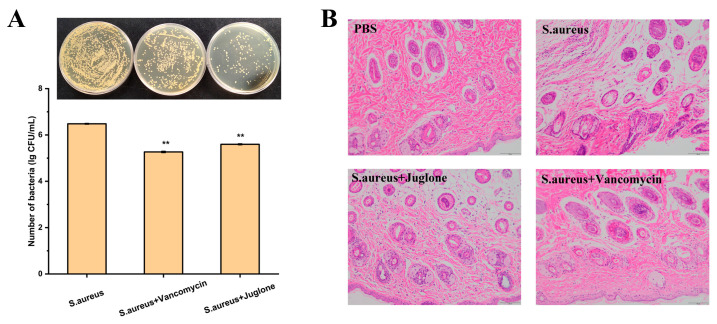
Effect of juglone topical treatment on wound bacterial load and wound histological analysis. (**A**) Bacterial load of wounds after 2 d of treatment in different treatment groups. (**B**) Histological evaluation of different treatment groups after 14 d of treatment. Wound tissue uninoculated with *S. aureus* (Control group); untreated wound tissue infected with *S. aureus* (*S. aureus* group); jugone-treated wound tissue with *S. aureus* contamination (*S. aureu* + juglone group); vancomycin-treated *S. aureus* infected wound tissue (*S. aureu* + vancomycin group). Scale bar was 100 μm. When *p* < 0.01, it is represented by **.

**Figure 10 ijms-24-03931-f010:**
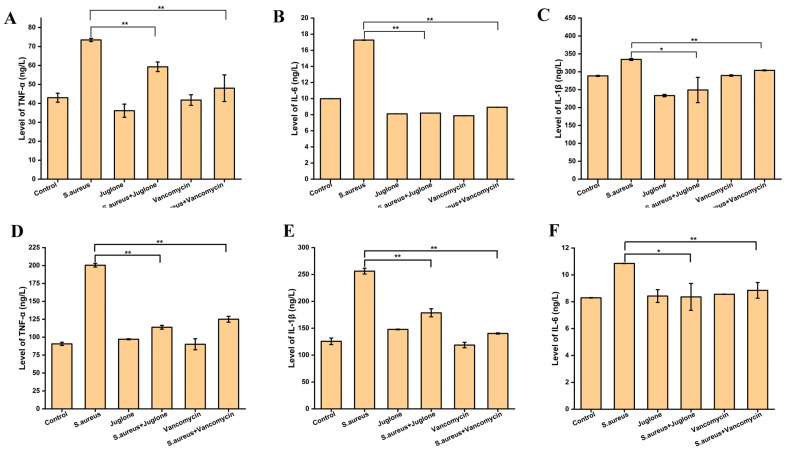
The effect of juglone on the levels of inflammatory factors in mice with *S. aureus*-infected wounds. The expression level of TNF-α (**A**), IL-6 (**B**) and IL-1β (**C**) in the serum of mice after three days of local treatment with juglone. The expression level of TNF-α (**D**), IL-1β (**E**) and IL-6 (**F**) in the wound tissue of mice after three days of local treatment with juglone. When *p* < 0.01, it is represented by **, when *p* < 0.05, it is represented by *.

**Figure 11 ijms-24-03931-f011:**
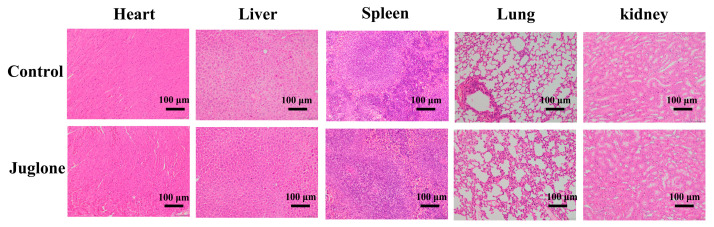
Safety assessment of juglone. The effect of juglone on the main organs (heart, liver, spleen, lung and kidney) of mice after 14 d of topical treatment. Scale bar was 100 μm.

**Table 1 ijms-24-03931-t001:** Minimum inhibitory concentration (MIC) and minimum bactericidal concentration (MBC) of juglone against *S. aureus*.

Strain	MIC (μg/mL)	MBC (μg/mL)	Original Source of Strain
ATCC29213 (MSSA)	1000	2000	American type culture collection
ATCC43300 (MRSA)	1000	2000	American type culture collection
ST97 (MRSA)	1000	2000	Cow mastitis wound

## Data Availability

Not applicable.

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
