# Peer review of "Antibacterial Activity of Juglone Revealed in a Wound Model of *Staphylococcus aureus* Infection"

_ijms, 2023, doi:10.3390/ijms24043931_

Round 1
Reviewer 1 Report
In the materials and methods part, there is no information how juglone was obtained. Was it a compound from the manufacturer or obtained by the authors.
In the references part, it is required to improve the writing of data to articles. Not all of these data are accurate.
Author Response
Response to Reviewer 1 Comments
Dear Reviewers and Editors,
We are very grateful to the reviewers and editors of the paper for their critical reading of the manuscript and many valuable recommendations for our further improvements. We have revised the manuscript and marked changes in blue in revised manuscript according to the comments.
Point 1: In the materials and methods part, there is no information how juglone was obtained. Was it a compound from the manufacturer or obtained by the authors.
Response 1: Thank you for your question. The juglone in the manuscript was purchased from Shanghai Yuanye Biotechnology Company, China. We have noted it in the "MIC and MBC" section of the materials and methods section. I'm sorry we didn't write it clearly.
Point 2: In the references part, it is required to improve the writing of data to articles. Not all of these data are accurate.
Response 2: Thank you for your question. We carefully checked the references and added the missing parts in the manuscript.

Reviewer 2 Report
The manuscript from Wan et al. describes the activity of juglone against S. aureus by evaluating some insights into its action mechanisms, antivirulence effects and in vivo properties in a wound model of Staphylococcus aureus infection. Although the MIC of juglone was high considering a isolated compound, the authors should that juglone at subinhibitory concentrations could inhibit key virulence factors of S. aureus.
The manuscript is interesting and informative. It deserves be published after some adjustments. It is important to note that due the journal style where the results section is presented before the methods section, some methodological details should be provided in the results to improve the understanding.
Please consider other important points:
- Please provide more details on the abstract file.
- Please indicate which strain was used in each assay in the results section.
- Please consider to provide the percentages of inhibition for each assay.
- The inflammatory mediators should also be measured at the wound tissue.
Other comments and suggestions are provided in the attached pdf file.

Author Response
Response to Reviewer 2 Comments
Dear Reviewers and Editors,
We are very grateful to the reviewers and editors of the paper for their critical reading of the manuscript and many valuable recommendations for our further improvements. We have revised the manuscript and marked changes in blue in revised manuscript according to the comments.
Point 1: Please provide more details on the abstract file.
Response 1: Thank you for your suggestion. We have added the information in the corresponding section to the abstract.
Point 2: Please indicate which strain was used in each assay in the results section.
Response 2: Thank you for your suggestion. We have added the strains used in each assay to the corresponding part of the manuscript.
Point 3: Please consider to provide the percentages of inhibition for each assay.
Response 3: Thank you for your suggestion. We have corrected the corresponding percentage in the manuscript.
Point 4: The inflammatory mediators should also be measured at the wound tissue.
Response 4: Thank you for your suggestion. In this experiment, we did ignore the problem you pointed out. We added the inflammatory mediators in the wound tissue of mouse skin in the manuscript. Details are shown in Figure 10.
Point 5: Line 14: Please express in micrograms per mL.
Response 5:Thank you for your suggestion. We have made corresponding modifications in the manuscript.
Point 6: Line 16: Please express in micrograms per mL.
Response 6:Thank you for your suggestion. We have made corresponding modifications in the manuscript.
Point 7: Line 17: Which mice type was used?
Response 7:Thank you for your question. The mice we used were Kunming mice, which we noted in the materials and methods section. It is also supplemented in the abstract.
Point 8: Line 17: What was the dose utilized?
Response 8:Thank you for your question. The dose we used in this part of the experiment is 50 μL juglone with a concentration of 1000 μg/mL (MIC).
Point 9: Line 18: Were the mediators quantification performed in wound tissue? Please make it clear.
Response 9:Thank you for your suggestion. In this experiment, we did ignore the problem you pointed out. We added the inflammatory mediators in the wound tissue of mouse skin in the manuscript. Details are shown in Figure 10.
Point 10: Line 26: Please amend this sentence as S. aureus can be part of the normal human microbiota.
Response 10:Thank you for your suggestion. We have corrected the expression.
Point 11: Line 64-65: Please provide the unit for MIC and MBC in micrograms per mL.
Response 11:Thank you for your suggestion. We have made corresponding modifications in the manuscript.
Point 12: Line 69: Please consider to provide the classification as MRSA and MSSA for each strain in the table 1.
Response 12:Thank you for your suggestion. We have added the MRSA and MSSA classification of each strain in Table 1.
Point 13: Line 69: Please consider to provide the full name instead the abbreviation in the table title.
Response 13:Thank you for your suggestion. We have made corresponding modifications in the manuscript.
Point 14: Please express using 1/2 x MIC, use this for other concentrations.
Response 14:Thank you for your suggestion. We have made corresponding modifications in the manuscript.
Point 15: Line 75: Please indicate which S. aureus strain was used in each assay.
Response 15:Thank you for your suggestion. We have added the S. aureus strains used in each assay to the manuscript.
Point 16: Line 83: Maybe would be better to represent these results using percentage of inhibition.
Response 16:Thank you for your suggestion. We have made modifications. Specifically, it can be seen from Figure 2A that when S. aureus ATCC29213 strain was treated with the MIC concentration of juglone, the membrane potential was significantly reduced by 31.10% compared with the control group (treated with the corresponding concentration of DMSO). Similarly, 2×MIC concentration of juglone also reduced the membrane potential of S. aureus to 67.44%.
Point 17: Line 87: Please consider to represent these results using percentage of inhibition.
Response 17:Thank you for your suggestion. We have made modifications. Specifically, As shown in Figure 2B, the MIC concentration of juglone significantly decreased the intracellular ATP concentration of S. aureus (P<0.01), which was 16.75%. When the concentration of juglone increased to 2×MIC, the intracellular ATP concentration of S. aureus decreased to 12.72%.
Point 18: Line 99: The authors could perform a quantitative assay to assess the ueffects of juglone in total proteins of S. aureus.
Response 18:Thank you for your suggestion. We quantified the total protein. It is supplemented in Figure 2D.
Point 19: Figure 4. Please consider to use white letters in the figures to improve the readability.
Response 19:Thank you for your suggestion. We have corrected the letters in Figure 4 to white.
Point 20: Line 154: Please represent these results using percentage of inhibition.
Response 20:Thank you for your suggestion. We have made corresponding modifications in the manuscript. Specifically, as shown in Figure 6A and Figure 6B, juglone significantly inhibited the hemolytic activity of S. aureus (P<0.01), which decreased the hemolytic activity to below 15.46%, and the inhibition was dose-dependent.
Point 21: Line 167: Please indicate which S. aureus strain was used in each assay.
Response 21:Thank you for your suggestion. We have added the strains used in each assay to the corresponding part of the manuscript.
Point 22: Line 203: The measurement of these inflammatory mediators should be also provided in the wound tissue.
Response 22:Thank you for your suggestion. In this experiment, we did ignore the problem you pointed out. We added the inflammatory mediators in the wound tissue of mouse skin in the manuscript. Details are shown in Figure 10.
Point 23: Line 232: Please provide the appropriate references in this paragraph where is needed.
Response 23:Thanks for your suggestions, we have supplemented the corresponding references to the manuscript.
Point 24: Line 232-248: Please provide the appropriate references in this paragraph where is needed.
Response 24:Thanks for your suggestions, we have supplemented the corresponding references to the manuscript.
Point 25: Line 535: Please indicate which volume was applied in the mice.
Response 25:Thank you for your question. The volume of juglone we used in mice is 50 μL. We have added it to the manuscript.
Other amendments have been added in the PDF file attached.

Reviewer 3 Report
Wan et. al., presented a very nice work where they showed that Juglone posses an antibacterial and wound healing potential against Staphylococcus aureus. I have few comments to strengthen the work.
1. In Figure 2C “Maker” should be replaced by “Markers”.
2. Figure 5B should contain the percentage inhibition in “Y axis” and write the full term instead of abbreviation.
3. I suggest using positive control in the in vitro experiments.
4. I suggest showing the cytotoxic effect of Juglone against primary cell lines.
Author Response
Response to Reviewer 3 Comments
Dear Reviewers and Editors,
We are very grateful to the reviewers and editors of the paper for their critical reading of the manuscript and many valuable recommendations for our further improvements. We have revised the manuscript and marked changes in blue in revised manuscript according to the comments.
Point 1: In Figure 2C “Maker” should be replaced by “Markers”.
Response 1: Thank you for your question. We have made corresponding modifications in the manuscript.
Point 2: Figure 5B should contain the percentage inhibition in “Y axis” and write the full term instead of abbreviation. S. aureus biofilm formation.
Response 2: Thank you for your question. We have made corresponding modifications in the manuscript.
Point 3: I suggest using positive control in the in vitro experiments.
Response 3: Thank you for your question. We have indeed ignored this issue. Tarek Zmantar et al. found in the study of “Use of juglone as antimicrobial and potential efflux pump inhibitors in Staphylococcus aureus isolated from the oral activity” that the MIC value of juglone and erythromycin to another standard strain 25923 of Staphylococcus aureus was consistent [1]. Jiayi Wang et al. also did not add positive control in the study of “Antimicrobial Activity of juglone against Staphylococcus aureus: From Apparent to Proteomic [2]”. At the same time, I added this question in the discussion part of the manuscript. Thank you again for making our manuscript more rigorous.
Point 4: I suggest showing the cytotoxic effect of Juglone against primary cell lines.
Response 4: Thank you for your question. We have indeed ignored this issue. Alexa-Maria Crotoru et al. in the study of “Novell Graphene Oxide/Quercetin and Graphene Oxide/Juglone Nanostructured Platforms as Effective Drug Delivery Systems with Biomedical Applications” used L929 fibrinoblast as the primary cell to prove that its juglone complex drug delivery system has good biocompatibility [3]. This proves the biosafety of juglone in primary cell experiment. In our manuscript, the biocompatibility of juglone was also proved by the method of experimental animal toxicity.
- Zmantar, T.; Miladi, H.; Kouidhi, B.; Chaabouni, Y.; Slama, R.B.; Bakhrouf, A.; Mahdouani, K.; Chaieb, K. Use of juglone as antibacterial and potential efflux pump inhibitors in Staphylococcus aureus isolated from the oral cavity. Microbial Pathogenesis 2016, 101, 44-49.
- Wang, J.; Cheng, Y.; Wu, R.; Jiang, D.; Bing, B.; Tan, D.; Yan, T.; Sun, X.; Zhang, Q.; Wu, Z. Antibacterial Activity of Juglone against Staphylococcus aureus: From Apparent to Proteomic. International Journal of Molecular Sciences 2016, 17, 965.
- Croitoru, A.M.; Morosan, A.; Tihauan, B.; Oprea, O.; Motelica, L.; Trusca, R.; Nicoara, A.I.; Popescu, R.C.; Savu, D.; Mihaiescu, D.E.; et al. Novel Graphene Oxide/Quercetin and Graphene Oxide/Juglone Nanostructured Platforms as Effective Drug Delivery Systems with Biomedical Applications. Nanomaterials 2022, 12, doi:10.3390/nano12111943.

Round 2
Reviewer 2 Report
The authors have improved the manuscript and it is suitable for publication in this current form.